# Investigating Primary Cilia during Peripheral Nervous System Formation

**DOI:** 10.3390/ijms22063176

**Published:** 2021-03-20

**Authors:** Elkhan Yusifov, Alexandre Dumoulin, Esther T. Stoeckli

**Affiliations:** Department of Molecular Life Sciences and Neuroscience Center Zurich, University of Zurich, Winterthurerstrasse 190, 8057 Zurich, Switzerland; elkhan.yusifov@uzh.ch (E.Y.); alexandre.dumoulin@mls.uzh.ch (A.D.)

**Keywords:** primary cilium, ciliogenesis, neural circuit formation, neural crest cells, DRG, boundary cap cells, sympathetic ganglia, PNS

## Abstract

The primary cilium plays a pivotal role during the embryonic development of vertebrates. It acts as a somatic signaling hub for specific pathways, such as Sonic Hedgehog signaling. In humans, mutations in genes that cause dysregulation of ciliogenesis or ciliary function lead to severe developmental disorders called ciliopathies. Beyond its role in early morphogenesis, growing evidence points towards an essential function of the primary cilium in neural circuit formation in the central nervous system. However, very little is known about a potential role in the formation of the peripheral nervous system. Here, we investigate the presence of the primary cilium in neural crest cells and their derivatives in the trunk of developing chicken embryos in vivo. We found that neural crest cells, sensory neurons, and boundary cap cells all bear a primary cilium during key stages of early peripheral nervous system formation. Moreover, we describe differences in the ciliation of neuronal cultures of different populations from the peripheral and central nervous systems. Our results offer a framework for further in vivo and in vitro investigations on specific roles that the primary cilium might play during peripheral nervous system formation.

## 1. Introduction

The primary cilium is a tiny nonmotile organelle that can be seen on most vertebrate cells [1,2,3]. It was discovered by the Swiss neuroscientist Karl Wilhelm Zimmermann in 1898 [4]. After its discovery, the primary cilium was neglected for a century as an evolutionary rudiment. However, studies in the last two decades have characterized the primary cilium as a signaling hub involved in distinct signaling cascades, such as Sonic Hedgehog (Shh) [5,6], Wnt [7,8], PDGFR [9,10], mTOR [11], and other pathways. Subsequent studies have demonstrated that the primary cilium is involved in the differentiation, proliferation, survival, polarity, and migration of cells [12]. That is why defects in the genes responsible for primary cilium formation and function have been identified as the cause for a group of disorders gathered under the umbrella of ciliopathies [13,14]. Common features in ciliopathies include retinal degeneration, craniofacial defects, polydactyly, intellectual disability, situs inversus, and cystic kidneys. As suggested by the name, proteins involved in ciliopathies are located either inside or at the base of the primary cilium [15]. Mutations in the genes responsible for primary cilia formation cause defective ciliary function in nonmotile ciliopathies and, thus, trigger various developmental and degenerative phenotypes in different tissues such as kidney, retina, and brain [16,17]. Human patients with ciliopathies, such as Joubert syndrome, can show pronounced abnormalities in the formation of the central nervous system (CNS). Translating observations from humans into model organisms has helped characterize pathological processes related to the loss of functional primary cilia at different levels, such as neuronal differentiation, migration, and axon tract formation [18,19,20]. However, to date, the potential role that the primary cilium might play during the formation of the peripheral nervous system is still poorly characterized, although several studies have suggested such a role [21,22].

Neural crest cells (NCCs) of the trunk are multipotent cells that delaminate from the dorsal neural tube and migrate laterally and ventrally using distinct trajectories. When they reach their final destination, they differentiate into dorsal root ganglia (DRG) neurons, sympathetic ganglia (SG) neurons, boundary cap cells (BCCs), or glial cells [23,24]. Neural crest cell development is, therefore, a prerequisite in the formation of a fully functional peripheral nervous system (PNS). Each step in the development of NCCs—delamination, migration, and differentiation into their derivatives—is temporally and transcriptionally regulated.

To learn more about the possible role of primary cilia in PNS development, it is crucial to first investigate whether distinct NCC populations possess a primary cilium during key stages of development in vivo. We could assess and confirm the presence of a primary cilium in migrating NCCs and their derivatives, such as DRG neurons, SG neurons, and BCCs in vivo. This confirmed that the primary cilium might be engaged in the development of the PNS at multiple levels. Moreover, we also characterized the timing of ciliation in dissociated and explant cultures of PNS and CNS neurons. We found that dissociated DRG neurons did not recover their primary cilium, whereas other neuronal populations, such as commissural neurons and motoneurons, largely did, albeit with distinct timing. Furthermore, when cultured as explants, neurons did not lose primary cilia in culture. Taken together, these findings will be useful for scientists trying to decipher molecular mechanisms of PNS assembly and the primary cilium in vivo and in vitro.

## 2. Results

### 2.1. Trunk Neural Crest Cells Bear a Primary Cilium during Their Early Migration

NCCs delaminating from the lumbar neural tube start migrating in a dorsal-to-ventral direction around Hamburger and Hamilton stage 16 (HH16) in chicken embryos and will then settle at their final destination prior to differentiation (Figure 1A–C) [25,26]. Immunohistochemistry was used to visualize the migration of NCCs and to characterize whether NCCs possess a primary cilium during their migration and early settlement in vivo. We stained transverse spinal cord cryosections for the NCC marker HNK1 [26] and the primary cilium marker Arl13B (Figure 1D–J) [27]. These multipotent stem cells bore a primary cilium during their early migration before arriving at the dorsal root level (white arrows, Figure 1E,F). Migrating NCCs reached the ventral roots at HH17 (Figure 1B,G) and the area lateral of the notochord at HH18 (Figure 1C,I). During migration and settlement, we found primary cilia located on the soma of NCCs (white arrows, Figure 1H,J). In fact, 88 ± 5% of HH18 migrating NCCs bore a primary cilium, with an average primary cilium length of 2.1 ± 0.5 µm (mean ± standard deviation, N(embryos) = 3; see also Figure 5 for a summary of the data on length and ciliation ratio).

Taken together, our data revealed that just after undergoing epithelial-to-mesenchymal transition and delamination, NCCs already bore a primary cilium and maintained it during their migration and early settlement.

### 2.2. Dorsal Root Ganglia Neurons Carry a Primary Cilium during Development

We then asked whether the primary cilium was maintained in NCC derivatives during development. We first focused on the sensory neurons located in the DRG. We found that the first Islet-1-positive, differentiated DRG neurons carried a primary cilium at HH18 (white arrows, Figure 2A). Interestingly, Islet-1-negative cells in the DRG also had a primary cilium, suggesting that either neuronal or Schwann cell precursors were ciliated (open arrow, Figure 2A). Certainly at HH26, 1E8-positive Schwann cell precursors located in DRG bore a primary cilium (arrows, Appendix A). At HH26 and HH30, cilia were evenly distributed throughout the DRG (white arrows, Figure 2B,C). We found that 86 ± 5% of Hoechst-positive cells carried a primary cilium in HH26 DRG. Given the fact that 75 ± 3% of these cells were Islet-1-positive (mean ± standard deviation, N(embryos) = 3, n(DRG) = 12), this suggested that the large majority of DRG neurons were ciliated at this stage (see also Figure 5). The average length of primary cilia in DRG was of 2.2 ± 0.5 µm (mean ± standard deviation, N(embryos) = 3, n(DRG) = 12). In line with these results, we made similar observations in mice at embryonic day E11.5, at the time when DRG afferents enter the spinal cord (white arrows, Figure 2D) [28].

We validated the staining of cilia obtained with antibodies against Arl13B with staining for the intraflagellar transport protein IFT88 (arrows, Appendix A) [29]. Furthermore, we utilized in ovo electroporation to transfect Arl13B-RFP in NCCs and their derivatives and could reveal the presence of primary cilia in Islet-1-positive DRG neurons (white arrow, Figure 2E).

Altogether, these results show that DRG neurons maintain their primary cilium when they polarize, send their axons towards the periphery and the CNS, and connect to the deeper layers of the spinal cord [30].

### 2.3. Boundary Cap Cells Bear a Primary Cilium at Both Ventral and Dorsal Roots

During the assembly of chicken trunk PNS, a subtype of migrating NCCs, the boundary cap cells (BCCs), will first stop and cluster at the ventral roots exit points (also termed motor axon exit points) of the spinal cord around HH18, as revealed by 1E8 immunostaining, recognizing the P0 protein (white open arrowheads, Figure 3A). Later, BCCs will do the same at the dorsal root entry zone level (white arrowheads, Figure 3A). BCCs are located at the interface between the CNS and the PNS. They play a role as gatekeepers between the CNS and the PNS, and they represent stem cells that later contribute to subtypes of sensory neurons, glial cells, and pericytes [31,32]. Using coimmunostaining of the BCC marker 1E8 and the primary cilium marker Arl13B, we were able to see that ventral and dorsal BCCs bore a primary cilium at all stages investigated in vivo (HH18–HH30; white arrows, Figure 3B–E and Appendix A). We quantified that 87 ± 5% of dorsal BCCs and 90 ± 3% of ventral BCCs were ciliated at HH26, with an average length of primary cilia of 2.8 ± 0.5 µm (dorsal BCC) and 3 ± 0.6 µm (ventral BCC; mean ± standard deviation, N(embryos) = 3; see also Figure 5). Moreover, we could confirm the ciliation of BCCs at HH26 with IFT88 staining (arrows, Appendix A).

Overall, our results indicate that BCCs bear a primary cilium throughout the early stages of development when neural circuits in the PNS and the CNS are formed.

### 2.4. Carrying a Primary Cilium Appears to Be a Common Feature on NCC Derivatives In Vivo

To further complement our in vivo characterization of ciliation, we also assessed two other populations of NCC derivatives: the sympathetic ganglia (SG) neurons and the melanocytes.

SG neurons are located in the SG chains and are part of the autonomic nervous system [33]. Coimmunostaining of the SG neuron markers Islet-1 or Tyrosine hydroxylase (TH), together with the primary cilia marker Arl13B, revealed that these neurons were ciliated at HH26 (white arrows, Figure 4A,B). We found that 96 ± 2% of SG neurons carried a primary cilium, with an average length of 2.5 ± 0.9 µm (mean ± standard deviation, N(embryos) = 3; Figure 5). The ciliation of these neurons was also confirmed by IFT88 staining of primary cilia (arrows, Appendix A).

A subset of trunk NCCs after delamination will migrate dorsolaterally and constitute the melanocyte lineage to ultimately form melanocytes in the skin [23,25]. We visualized melanocytes in the skin of the trunk at HH26 using the MelEM antibody and could see that these cells bore a primary cilium, with Arl13B costaining (white arrows, Appendix A). In fact, 93 ± 4% of these cells bore a primary cilium, with an average length of 1.6 ± 0.6 µm (mean ± standard deviation, N(embryos) = 3; Figure 5).

Collectively, our results showed that both SG neurons during their early development and melanocytes located in the skin bore a primary cilium.

Taken together, our qualitative and quantitative assessments of ciliation of neural crest cells and their derivatives demonstrate that they all bear a primary cilium during development. Moreover, the percentage of ciliation is really high, with around 90% of cells in each population carrying a cilium. There is some variability in ciliary length within each cell population and also between cell types. However, it is not clear to what extent these differences reflect variability in length measurements due to ciliary orientation.

### 2.5. Cultured Neurons Reveal Heterogeneity in Their Ciliation

Since in vitro studies using dissociated neurons and neural explants have been widely used to investigate molecular mechanisms during neurodevelopment [34], it is important to assess whether primary cilia persist in these cultures.

We first used HH30 DRG primary neurons and cultured them for 1, 2, 5, or 7 days in vitro (DIV). We stained them for the neuronal DRG markers Islet-1 and Arl13B to reveal primary cilia (Figure 6A–D). We rarely observed Arl13B-positive cilia in these cultures. In fact, quantifications indicated that the average ciliation rate of DRG neurons was invariably around 20% at all culture times (Figure 6F and Table 1; *p* ≥ 0.05). Noteworthy, the large majority of Schwann cells located in DRG, which were cocultured with DRG neurons and costained with 1E8 antibodies, carried a primary cilium already after 1DIV (white arrowhead, Figure 6E). Quantifications showed that between 80% and 90% of them carried a primary cilium after 1, 2, 5, or 7 DIV without any significant difference between culture times (Figure 6G and Table 1; *p* ≥ 0.05). This suggested that the culture conditions were not incompatible with ciliogenesis or cilia maintenance. We then examined the ciliation of HH30 DRG neurons cultured as explants after 1 DIV. In contrast to the culture of dissociated cells, we found that the majority of DRG neurons bore a primary cilium (white arrows, Figure 6H). On average, 83 ± 2% of Hoechst-positive cells had a primary cilium. Of those cells, 93 ± 1% were Islet-1-positive DRG neurons (mean ± standard deviation, n = 12 explants, N = 3 replicates).

As DRG neurons belong to the PNS and cannot consistently regain a primary cilium when cultured as dissociated neurons, we assessed and compared the ciliation rate of CNS neurons in culture. As seen for DRG neurons in vivo, spinal cord neurons carried a cilium in vivo at HH26. We found a primary cilium on most of the dorsally-located contactin-2/axonin-1-positive commissural neurons and the ventrally-located Islet-1-positive motoneurons (white arrows, Figure 7A,C). Robust ciliation was still observed in both motoneurons and commissural neurons cultured as explants (white arrows, Figure 7B,D). Interestingly, the ciliation of dissociated motoneurons was around 50% after 1 and 2 DIV, then it significantly increased up to around 80% after 5 DIV and stayed at a similar level after 7 DIV (Figure 7G and Table 1; *p* < 0.0001). In contrast, the majority (~66%) of dissociated commissural neurons carried a primary cilium already after 1 DIV and maintained it at a similar level up to 7 DIV (Figure 7H and Table 1; *p* ≥ 0.05).

Taken together, our culture experiments showed that DRG neurons are only robustly ciliated when cultured as explants. When cultured as dissociated cells, they lost the primary cilium and were unable to regain it. This is in contrast to neuronal populations from the CNS, such as motoneurons and commissural neurons of the spinal cord, where cilia were maintained or regained after some days in dissociated cultures.

## 3. Discussion

In this study, we analyzed the presence of primary cilia in different cell populations of developing chicken PNS, between HH16 and HH30 in vivo. Our data showed that NCCs and their derivatives carried a primary cilium during the early and later stages of PNS formation (Figure 1, Figure 2, Figure 3, Figure 4 and Figure 5, Appendix A).

We could see that migrating NCCs carried a primary cilium while migrating to their final locations, where they settle and differentiate (Figure 1). This observation suggests that already at a very early stage of trunk PNS formation, the primary cilium might potentially play a role in the migration and correct settlement of the NCCs. Interestingly, in human ciliopathies, patients have very often an abnormal development of the craniofacial complex, which suggests aberrant cranial NCC migration [35]. In line with this, studies using animal models of ciliopathies suggest that cranial NCC migration and development are impaired in a cell-autonomous manner when primary cilia are compromised. A lack of a functional primary cilium led to defects in the development of several organs, such as the heart, cranial structures, the tongue, and the submandibular gland [36,37,38,39,40,41,42]. Interestingly, NCC development at the trunk level involves signaling pathways, such as Wnt, Sonic Hedgehog, TGFβ, or chemokine signaling, which are linked to the primary cilium or have components that have been shown to localize in it in other cell types [43,44,45,46,47]. Thus, a cell-autonomous role for primary cilia in early NCC development in the trunk is very likely. In line with this hypothesis, previous results have suggested a role for the primary cilium in trunk NCC differentiation in mice.

Secondly, we described the ciliation of differentiated sensory neurons within the DRG (Figure 2); DRG neurons connect the periphery to the CNS. According to our findings, they were ciliated at all the key stages: when they polarize (HH18), sending central afferents into the CNS and peripheral axons toward the periphery (HH26), and when they produce collaterals growing within the deeper layers of the spinal cord (HH30, Figure 2A–C, respectively) [48,49]. In adult mice, DRG neurons were shown to bear a primary cilium, suggesting that they might maintain it after the PNS is formed [22]. Interestingly, silencing of the Joubert syndrome protein C5orf42 (also termed CPLANE1 or Jbts17)—a protein required for ciliogenesis—in chicken NCCs and neural tube led to obvious defects in the development of DRG central afferents and the sciatic nerve, suggesting that a functional primary cilium might play a role in the development of DRG axons at both the periphery and central afferent levels [21]. Furthermore, the loss of Bardet–Biedl syndrome proteins in mice was shown to lead to an alteration in sensory innervation in the skin, suggesting an abnormal development of peripheral sensory axons in ciliopathies [22]. Therefore, our results will be useful for further investigating the possible roles of a functional primary cilium in sensory neuron axonogenesis, axon guidance, and axon branching in vivo, given the fact that the DRG system is a very accessible system to study each one of these steps.

Another important population of NCC derivatives that we analyzed was the BCCs. BCCs bore a primary cilium at both the ventral and dorsal roots as soon as they clustered and maintained their cilium, at least until HH30 (Figure 3 and Appendix A). Several studies have demonstrated a role for these cells as a “gatekeeper” at the CNS–PNS boundary at the ventral root level by maintaining the soma of motoneurons in the ventral spinal cord by Semaphorin 6A-mediated repulsive signals [31,50,51]. In the dorsal roots, they might play a role in orchestrating the correct guidance of DRG central afferents into the spinal cord in higher vertebrates [52]. Interestingly, after C5orf42 knockdown in neural crest cells and neural tube, dorsal root formation and DRG patterning were aberrant [21]. It is possible that such phenotypes result from defects in the clustering of these cells at the dorsal root entry zone. Further experiments will be needed to verify this possibility. BCCs also serve as stem cell niches, and some of them keep proliferating after clustering [52]. In fact, BCCs contribute to subtypes of sensory neurons, Schwann cell precursors, and pericytes [32]. Therefore, it will be interesting to study in detail whether proliferating BCCs bear a primary cilium and whether a functional primary cilium would be required for their proliferation, as was shown for granule neuron progenitors of the developing cerebellum [53].

Additionally, to complement these results, we could also detect primary cilia on SG neurons and melanocytes at early stages (Figure 4 and Appendix A). As for DRG neurons, the presence of the primary cilium during the development of SG neurons might be of interest, although no hints at a possible role during their development have been reported yet. The presence of primary cilia in melanocytes is in line with observations made in the zebrafish [54].

Finally, we assessed the ciliation of different populations of neurons in vitro. We found that the widely used dissociated embryonic DRG neurons did not consistently recover their primary cilium in culture even after 7 DIV (Figure 6E–G). Only around 20% of DRG neurons carried an Arl13B-positive cilium. This was in contrast with previous data showing ciliation of P0 mouse hippocampal neurons after 7 DIV (Figure 6E–G) [55]. However, the majority of DRG neurons cultured as explants for 1 DIV was ciliated (Figure 6H). DRG neurons have been widely used to decipher molecular mechanisms of axonal outgrowth, axon guidance, and branching in vitro [56,57,58]. Interestingly, some of the major guidance receptors, such as Robo1 and Neuropilin-1, have recently been localized to the primary cilium of neurons in the CNS and mouse fibroblasts, respectively [43,59]. Hence, it is possible that some of the signaling pathways involved in axonal development might be transduced at the primary cilium level. Our results stress the fact that for in vitro studies investigating signaling pathways that might involve the primary cilium, a proper culture system should be carefully chosen. In the case of embryonic DRG neurons, DRG explants rather than dissociated cultures should be preferentially used.

Furthermore, we could also detect heterogeneity in the ciliation of CNS neurons compared to DRG neurons. The majority of commissural neurons and motoneurons carried a primary cilium when cultured as explants, as was the case for DRG neurons (Figure 7A–D). Dissociated commissural neurons were robustly ciliated already after 1 DIV, but for motoneurons, only after 5 DIV (Figure 7E–H). This suggests differences in the capacity for ciliogenesis in vitro among these neuronal cell types once they are dissociated: from being fast (commissural neurons) to slow (motoneurons) to inefficient (DRG neurons) in rebuilding a primary cilium. These differences might be due to changes in the expression of genes involved in ciliogenesis in a specific population of neurons compared to others [60,61]. The fact that only 50% of motoneurons can recover a primary cilium after 2 DIV might reflect a subpopulation-dependent difference. Further experiments using specific markers for specific subpopulations of motoneurons will be required to investigate this possibility. Moreover, there might be a general difference in the capacity of reciliation between CNS neurons and PNS neurons. Noteworthy, DRG neurons, under certain conditions, are able the regenerate their peripheral and central axons in vivo, whereas CNS neurons are not [62]. Therefore, it would be interesting to investigate the ciliation in DRG during axon regeneration as there might be some links between signaling cascades involved in regenerative responses and the primary cilium, as recently reported in retinal ganglion neurons [63].

Overall, the presence of primary cilia in NCCs and all their derivatives studied above in chicken embryos in vivo prompts further research to explore the role that the primary cilium might play during PNS formation. In particular, in ovo RNAi-based knockdown or CRISPR/Cas9-based knockout of specific genes will help to address these questions in more detail in chickens [64,65,66,67]. Our results pave the way for further studies to utilize chicken embryos as a model for deciphering the potential role of the primary cilium in neural circuit formation in the PNS.

## 4. Materials and Methods

### 4.1. Embryo Dissection and Fixation

The developmental stages of the chicken embryos were determined according to Hamburger and Hamilton [68]. The bodies were pinned down with insect pins (stainless steel pins) in a silicon-layered dish filled with cold PBS. The internal organs were removed until the vertebrae and the ribs were visible. For fixation, the extremities of older embryos (HH30) were shortened. Embryos were transferred to a new dish and fixed in 4% paraformaldehyde in PBS at room temperature for 1 h (HH18-HH26) or 2 h (HH30). Then, they were washed 3 times for 10 min with PBS at RT. For cryoprotection of the tissue, embryos were transferred to 25% sucrose in PBS and incubated for at least 24 h at 4 °C. For cryosectioning, the tissue was embedded in OCT compound (Tissue-Tek, Sakura Finetek, Alphen aan den Rijn, Netherlands), frozen, and cut with a thickness of 25 μm with a cryostat (Leica CM1850, Muttenz, Switzerland).

### 4.2. In Ovo Electroporation

In ovo electroporation was used to target neural crest cells in HH12–14 embryos, as described previously [69,70]. A DNA mix containing 25 ng/µL β-actin::hrGFPII and 25 ng/µL β-actin::Arl13B-RFP plasmids [71], diluted in PBS and 0.1% Fast Green, was injected into the central canal of the neural tube and unilaterally electroporated using a BTX ECM830 square-wave electroporator (five pulses at 18 V, with 50 ms duration each and a 1-s interpulse interval; BTX, Holliston, MA, USA), as previously described [70]. Eggs were then incubated for 3 more days at 39 °C until embryos reached stage HH26.

### 4.3. Immunohistochemistry

To prevent unspecific antibody binding, transverse spinal cord sections were incubated in blocking buffer (5% FCS in PBS) containing 0.25% Triton X-100 for one hour in a humid chamber. They were then washed 3× for 10 min each with PBS containing 0.25% Triton X-100 at room temperature. Primary antibody mixtures were prepared in blocking buffer and added to the slides (Table 2). Slides were incubated overnight at 4 °C in a humid chamber.

The next day, the spinal cord sections were washed 3× for 10 min each at room temperature with PBS/0.25% Triton X-100. Afterward, secondary antibodies diluted in blocking buffer were added to each slide (Table 3). The slides were then incubated for 2 h in a humid chamber at room temperature, protected from light. Before mounting, the sections were counterstained with Hoechst (Invitrogen (Thermo Fisher Scientific, Waltham, MA, USA) catalog number H3570, 2.5 µg/mL in PBS) for 10 min at room temperature, followed by 3 × 10 min each washing with PBS/0.25% Triton X-100 and 2× for 5 min each with PBS at room temperature. Afterward, the slides were either mounted with Mowiol/DABCO or further stained for primary cilia with the protocol outlined below.

### 4.4. Immunohistochemistry for Primary Cilia

To reduce the nonciliary staining of Arl13b or IFT88 antibodies and to get clear ciliary staining, we had to adapt and optimize standard protocols. Reducing the incubation time with the primary antibody greatly increased the ratio of the signal in the cilia versus the signal outside the cilia. Importantly, to stain primary cilia in both CNS and PNS, no (additional) detergent was used. Primary cilia were stained with rabbit anti-Arl13B antibody or rabbit anti-IFT88 diluted at 1:1000 in 5% FCS in PBS for 2 h at room temperature (Table 2). Following this incubation, the slides were washed 3 times with PBS for 10 min each, and a secondary antibody (donkey-anti-rabbit-Cy3), diluted at 1:1000 in blocking buffer, was given to slides for 2 h at room temperature (Table 3). Finally, the sections were washed 3 × 10 min each with PBS and mounted as described above. Note that the cilia staining protocol was always performed after immunostaining of the marker of interest (see Materials and Methods Section 4.3); thus, the tissue was already permeabilized with Triton-X-100 detergent.

### 4.5. Cultures of Dissociated Neurons

For accessing the DRG, HH30 embryos were pinned down, as described for immunohistochemistry, and the ventral vertebrae and the spinal cord were removed to access the DRG. For motoneurons and commissural neurons, open-book preparations of spinal cords dissected from HH26 embryos were used, and stripes of tissue were cut at the ventral region (motor column) or the most dorsal region (commissural neurons) of the spinal cord with small spring scissors [70]. Each population was dissociated by a 20-min incubation at 37 °C in 0.25% Trypsin in PBS (Invitrogen, cat# 15090-046) containing DNase (Roche (Basel, Switzerland), cat# 101 041 590 01; final concentration: 0.2%), followed by pelleting by centrifugation (5 min, 1000 rpm, room temperature) and trituration with a fire-polished Pasteur pipette in culture media. Finally, cells were resuspended in the respective culture medium (Table 4). N3 supplement was added to a final concentration of 100 μg/mL transferrin, 10 μg/mL insulin, 20 ng/mL triiodothyronine, 40 nM progesterone, 200 ng/mL corticosterone, 200 μM putrescine, and 60 nM sodium selenite (all from Sigma, Buchs, Swizerland). Penicillin and streptomycin (Invitrogen, cat# 15140-122) were added to the media for a final concentration of 100 units/mL and 100 μg/mL, respectively. Cells were plated in 8-well Lab-Tek plates (20,000 cells per well; Nunc (Thermo Fisher Scientific, Waltham, MA, USA), cat# 177445) and cultured for the desired time at 37 °C with 5% CO_2_. Lab-Tek plates were precoated with poly-L-lysine (20 μg/mL, Sigma, cat# P-12374) and coated with 10 μg/mL laminin (Invitrogen, cat# 23017-015). When neurons were cultured for more than 2 days, fresh medium was added every two days (half the volume of the old medium in the well was exchanged with a fresh one).

Cells were fixed after 1, 2, 5, or 7 DIV by adding one volume of prewarmed 4% PFA in PBS to each well and incubation for 5 min at 37 °C with 5% CO_2_. Then, the medium-PFA mix was exchanged for prewarmed 4% PFA in PBS, and cells were incubated for another 15 min with the same conditions before being washed with PBS at room temperature.

### 4.6. Explants Cultures

For explant cultures, LabTek wells were coated with poly-L-lysine and laminin as described above. Explants of DRG, motoneurons, or commissural neurons were transferred directly into LabTek wells containing a total of 400 μL medium (Table 4). After 1 DIV, 400 μL of prewarmed 2% PFA, with 15% sucrose in PBS (fixation buffer), was added to gently fix the explants, as described previously [72]. Following 40 min of incubation at room temperature, the upper phase of the solution (400 μL) was discarded, and 400 μL of fresh fixation buffer was given. After incubation for another hour, the explants were washed 3 × 10 min each with PBS at room temperature.

### 4.7. Immunocytochemistry

Cells were permeabilized with 0.1% Triton-X 100 in PBS (for dissociated neurons) or 0.25% Triton-X 100 in PBS (for explants) for 4 min at room temperature. They were rinsed 3× for 5 min each with PBS and blocked 15 min with 5% FCS in PBS (blocking buffer) at room temperature. Primary antibodies were diluted in blocking buffer and added to the cells for 1 h at room temperature (Table 2). Cells were washed again with PBS and incubated in secondary antibodies diluted in blocking buffer for 1 h at room temperature (Table 3). Cells were counterstained with Hoechst (2.5 µg/mL in PBS, cat# H3570) for 4 min at RT and rinsed 3 × 5 min each with PBS before mounting, as described above.

### 4.8. Microscopy

Images were taken with an Olympus BX61 upright microscope equipped with a spinning disk unit (BX-DSU, Olympus, Tokyo, Japan) and a 10× air objective (UPLFL PH 10×/0.30, Olympus), a 40× water objective (UAPO W/340 40×/1.15, Olympus), or a 60× oil objective (PLAPON O 60×/1.42, Olympus) and an Orca-R^2^ camera (Hamamatsu, Shizuoka, Japan) with Olympus CellSens Dimension 2.2 software. Cilia were visualized with 60× or 40× objectives and images were acquired with 20 planes for 40× images, with a 0.32-µm distance between planes. For 10× and 60× images, 3.54 and 0.24 µm distances were applied, respectively. The number of planes was variable. High magnification images were 2D deconvolved (nearest neighbor) using Olympus CellSens Dimension 2.2 software. Maximum intensity z-projection was created with the same software, and all images were analyzed and equally modified in Fiji/ImageJ [73].

### 4.9. Quantification of Ciliation Rate and Cilia Length

To quantify the number of primary cilia in HH26 DRG in vivo and HH30 explants in vitro, Fiji/ImageJ was used with images acquired with the 40× objective. First, a 2-channel picture with Islet-1 (green) and Arl13B (red) was split into separate channels (Figure 8A). On the picture displaying the primary cilia (Arl13B, red), the “triangle” threshold was applied (Figure 8B). The picture showing the Islet-1-positive DRG neurons was used to select the region of interest of the DRG with the freehand selection tool (Figure 8C). This selection was transferred to the thresholded primary cilia picture (Figure 8D). Finally, cilia were automatically counted within the region of interest using the “count particles” option (0–1 circularity, 0.2 µm^2^ minimum size; Figure 8E). The number of Hoechst-positive and Islet-1-positive cells in the region of interest was counted manually to calculate the percentage of ciliation within each DRG. To quantify the percentage of ciliation in dissociated cultures, the numbers of total and ciliated neurons/cells per picture were manually counted by an experimenter, blind to the experimental conditions, using Fiji/ImageJ. The percentage of ciliation per replicate was calculated as the average from 4 adjacent pictures taken in the center of the well. For other populations, like migrating NCCs, BCCs, SG neurons, and melanocytes, it was easier to identify the ciliated cell with certainty, and, therefore, it was possible to calculate the ciliation rates directly. The ciliary length was manually measured in maximum projections of a Z-stack using the tracing tool in ImageJ in all populations mentioned in the text. It is, therefore, possible that the average cilia length might be underestimated as primary cilia oriented in the *Z*-axis of the stack would appear shorter than what they really are.

### 4.10. Statistical Analysis

Statistical analyses were carried out using GraphPad Prism 8 software. All data were assessed for normality (normal distribution) using the D’Agostino and Pearson omnibus K2 normality test and visual assessment of the normal quantile–quantile plot before choosing an appropriate (parametric or nonparametric) statistical test. All tests used in this study are mentioned either in the text or in the legend of figures.

## Figures and Tables

**Figure 1 ijms-22-03176-f001:**
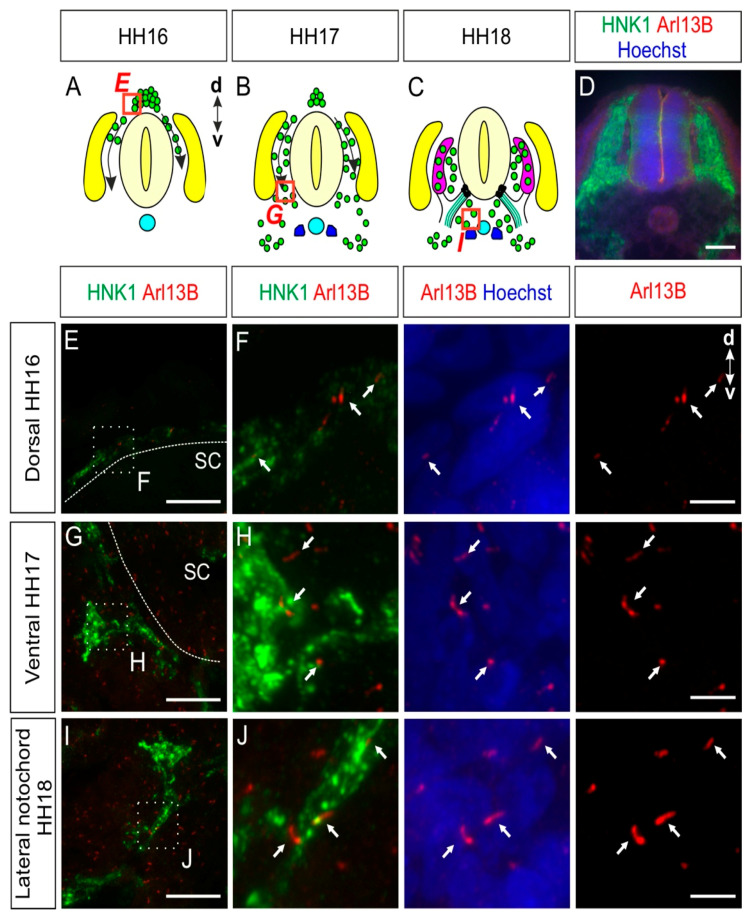
Trunk neural crest cells (NCCs) possess a primary cilium during early migration and settlement. (**A**–**C**) Trunk NCCs migrate ventrally between HH16 and HH18 on both sides of the neural tube. Green: NCCs, turquoise: notochord, purple: dorsal root ganglia (DRG), yellow: dermomyotome, blue: sympathetic ganglia chain, black dots: boundary cap cells (BCCs), green lines: ventral roots. d, dorsal; v, ventral. (**D**) A transverse section of the spinal cord of an HH18 embryo stained with HNK1 to stain NCCs (green) and with Arl13B antibodies to visualize primary cilia (red). Nuclei were counterstained with Hoechst (blue). (**E**,**F**) NCCs were visualized at HH16 before arriving at the dorsal root level. They were found to bear a primary cilium (white arrows). (**G**–**J**) NCCs located at the ventral root level (**G**) and the area lateral to the notochord (**I**) possessed a primary cilium (white arrows). Squares with dashed lines represent the region of interest shown in the right panels. Dashed lines represent the boundary of the spinal cord. d, dorsal; v, ventral; SC, spinal cord. Scale bars: 100 (**D**), 25 (**E**,**G**,**I**), and 5 μm (**F**,**H**,**J**).

**Figure 2 ijms-22-03176-f002:**
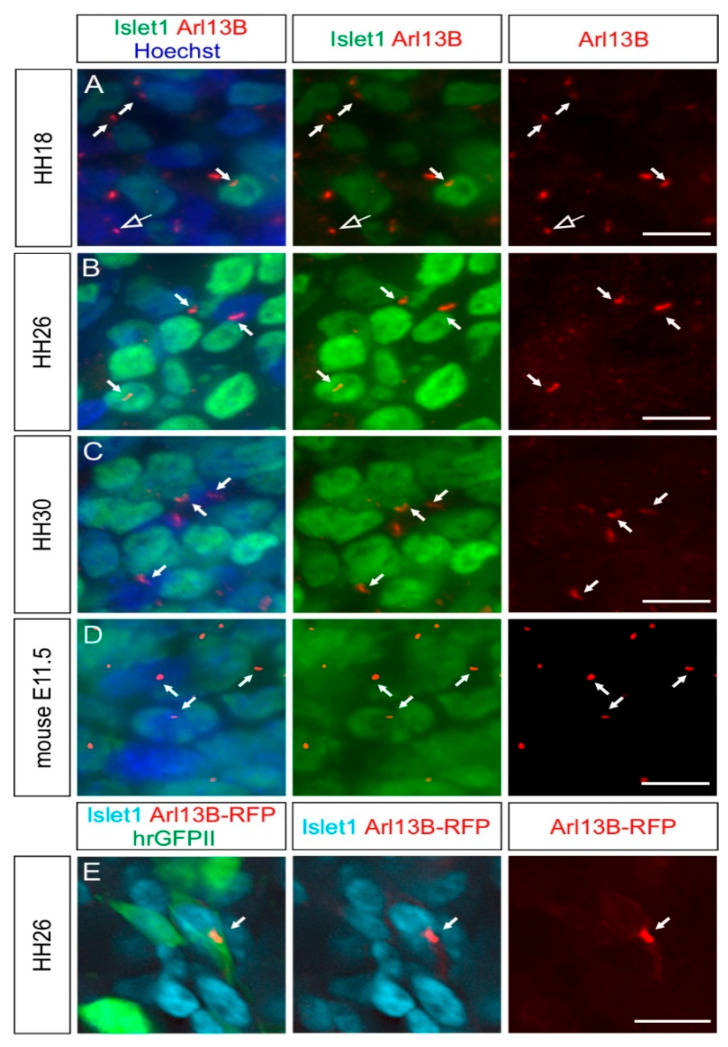
Embryonic DRG neurons possess a primary cilium in chicken and mouse embryos. (**A**–**D**) Micrographs of Islet1-positive DRG neurons (green) bearing an Arl13B-positive primary cilium (red, white arrows) at stage HH18, HH26, and HH30. The open arrow points to a primary cilium on an Islet-1-negative cell. (**D**) E11.5 mouse DRG neurons also possessed a primary cilium (white arrows). Nuclei were counterstained with Hoechst (blue). (**E**) Primary cilia were visible after RFP staining (red) of HH26 Islet-1-positive DRG neurons (cyan) after coelectroporation of Arl13B-RFP (white arrow) with hrGFPII (green). Dorsal is up. Scale bars: 10 (**A**–**D**) and 5 μm (**E**).

**Figure 3 ijms-22-03176-f003:**
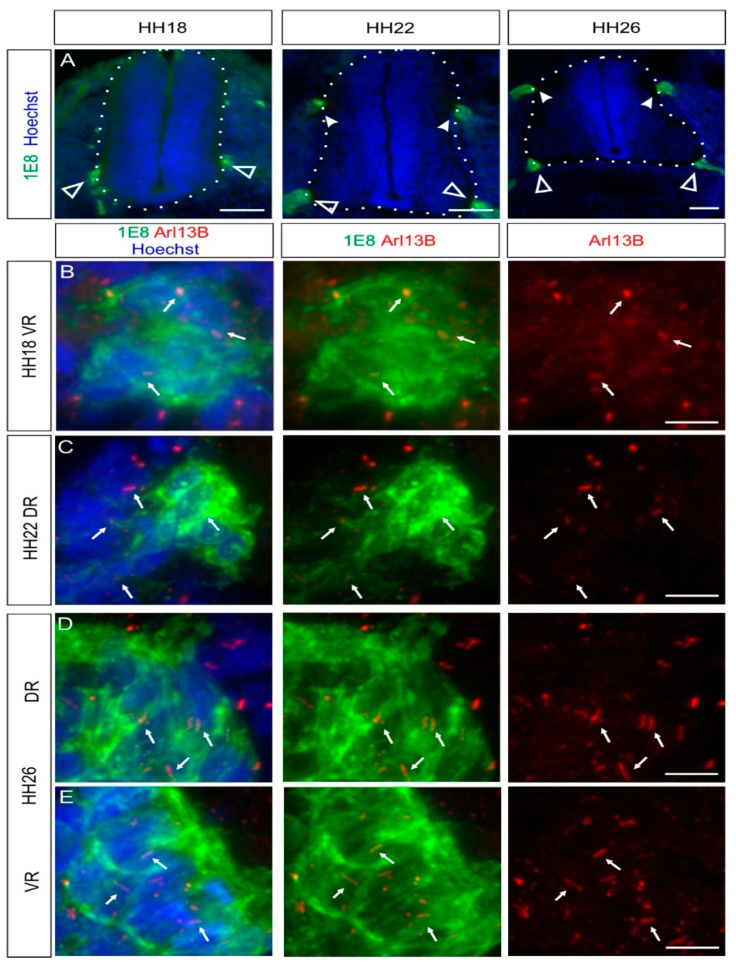
BCCs possess a primary cilium during the early development of the central nervous system (CNS)–peripheral nervous system (PNS) boundary. BCCs were stained with 1E8 antibodies (green) and primary cilia with Arl13B antibodies (red). (**A**) BCC clusters were initially detected at the ventral roots at HH18 (white open arrowheads). BCC clusters at the dorsal roots were observed at HH22 (white arrowheads). Nuclei were counterstained with Hoechst (blue). (**B**–**E**) BCCs were ciliated at both dorsal and ventral roots between HH18 and HH26 (white arrows). White dashed lines represent the boundary of the spinal cord (CNS). DR, dorsal roots; VR, ventral roots. Dorsal is up. Scale bars: 100 (**A**) and 10 μm (**B**–**E**).

**Figure 4 ijms-22-03176-f004:**
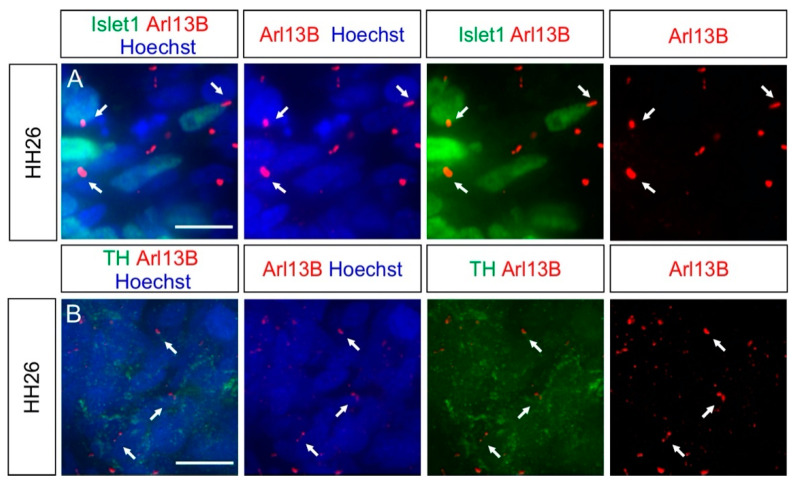
Other NCC derivatives, such as sympathetic ganglia (SG) neurons, possess a primary cilium. Islet-1 (**A**) and TH (**B**) antibodies were used to stain HH26 SG neurons (green). (**A**,**B**) Arl13B staining revealed that these neurons were ciliated (white arrows). Nuclei were counterstained with Hoechst (blue). Dorsal is up. TH, tyrosine hydroxylase. Scale bars: 10 μm.

**Figure 5 ijms-22-03176-f005:**
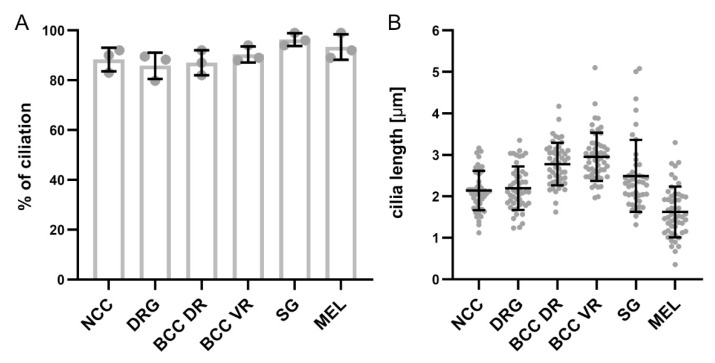
Quantification of the percentage of ciliated cells and ciliary length in NCCs and their derivatives. (**A**) Graph showing the average percentage of ciliation in migrating NCCs at HH18 and NCC derivatives at HH26. Note that for DRG, the average percentage of Hoechst-positive cells bearing a cilium is shown. (**B**) Graph showing the average cilia length in migrating NCCs at HH18 and NCC derivatives at HH26. Error bars represent standard deviation. Data points represent the average percentage for each embryo (**A**) and single data points of cilia length that were measured (**B**), respectively. DR, dorsal roots; VR, ventral roots, MEL, melanocytes.

**Figure 6 ijms-22-03176-f006:**
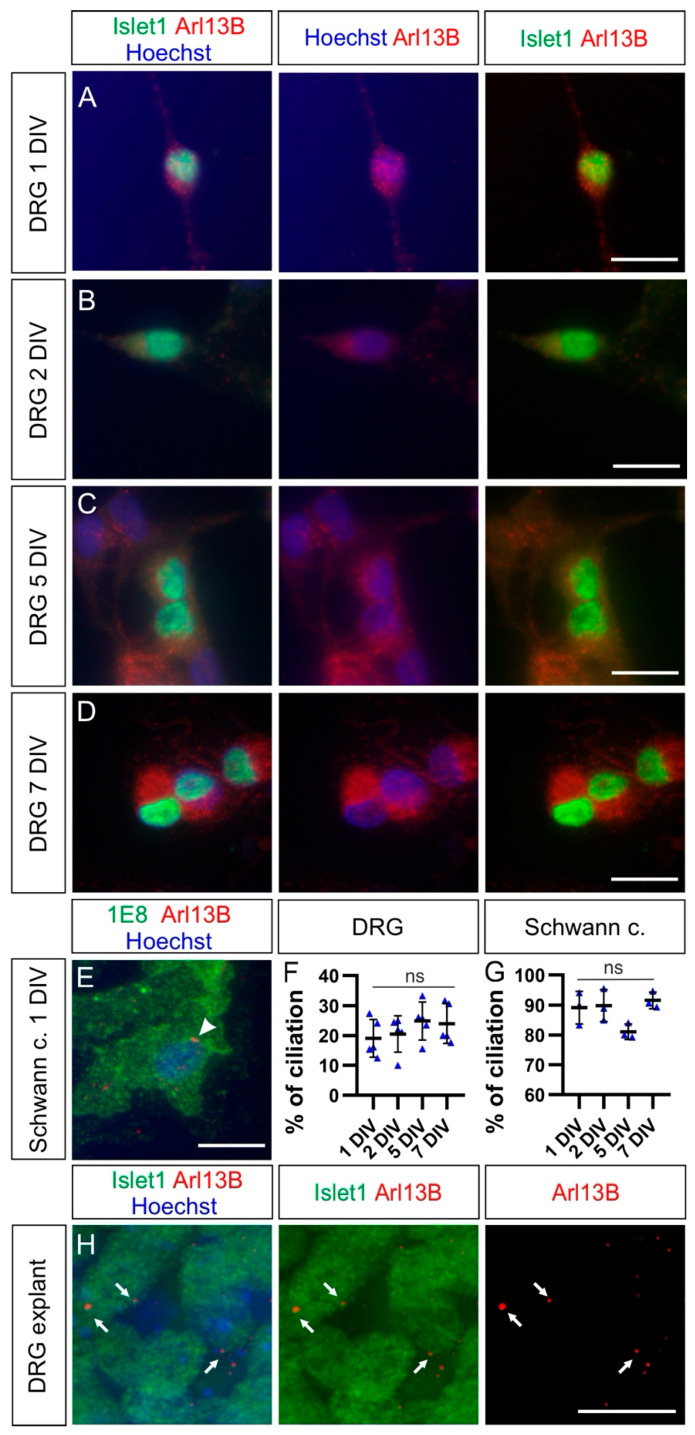
DRG neurons bear a primary cilium in explants but not in dissociated cultures. (**A**–**D**) Dissociated DRG neurons cultured for 1, 2, 5, or 7 days in vitro (DIV) were stained for Islet1 (green) together with Arl13B (red), revealing primary cilia, and counterstained with Hoechst (blue) to visualization nuclei. No clear Arl13B-positive ciliary structures could be seen on these cells. (**E**) Most Schwann cells possessed a primary cilium in dissociated cell culture (white arrowhead). (**F**,**G**) Quantification of the average percentage of ciliation in dissociated cultures of DRG neurons and Schwann cells up to one week of culture. Error bars represent standard deviations. Kruskal–Wallis test with Dunn’s multiple-comparisons test (**F**) and one-way ANOVA with Dunn’s multiple-comparisons test (**G**). *p* ≥ 0.05, ns. (**H**) Islet1-positive DRG neurons in DRG explants possessed a primary cilium (red) after 1 DIV (white arrows). Schwann c., Schwann cells. Scale bars: 15 μm.

**Figure 7 ijms-22-03176-f007:**
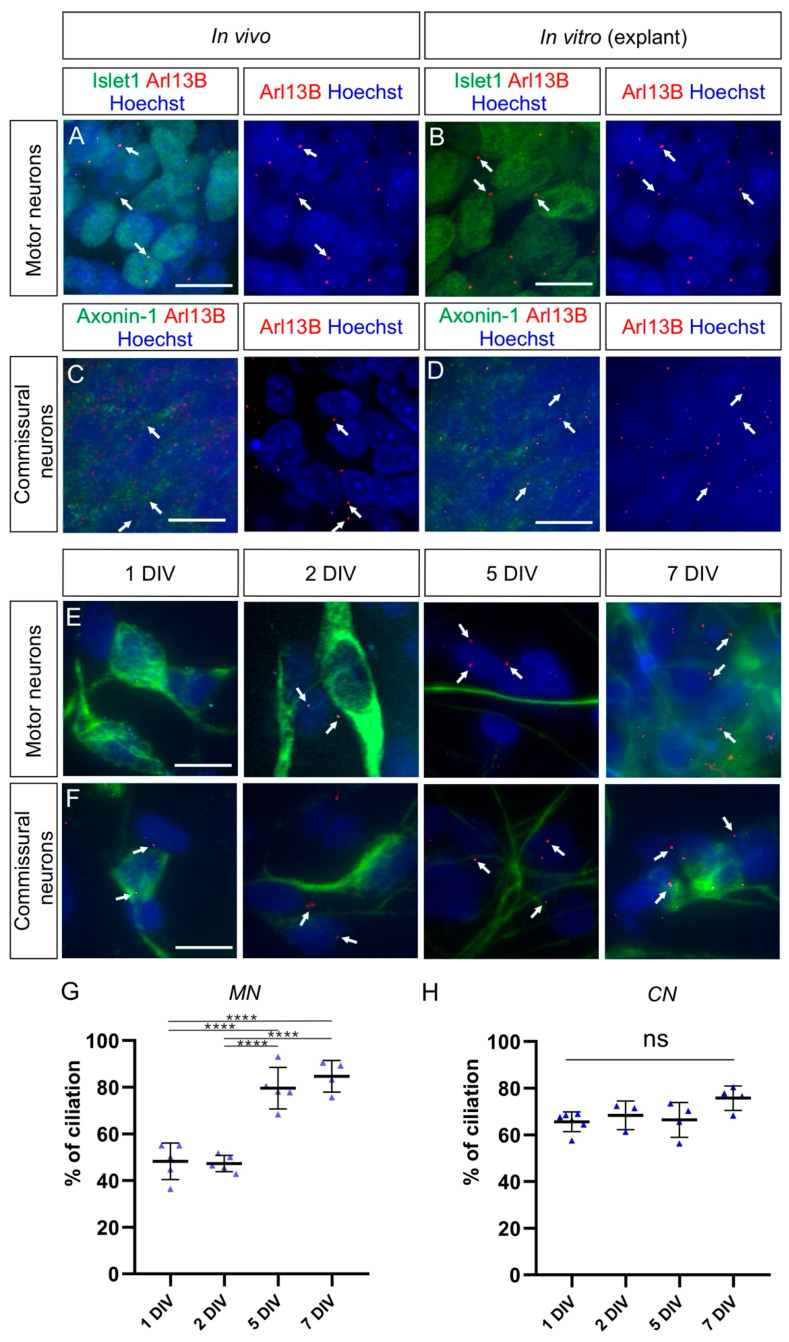
Commissural neurons and motor neurons possess a primary cilium in vivo and in explants cultures, but the ciliation rate is distinct for each population in dissociated cell cultures. (**A**) Islet-1-positive motoneurons located in the ventral horn of the spinal cord bore a primary cilium at HH26 in vivo (white arrows). (**B**) Motoneurons also possessed a primary cilium in explants cultured for 1 DIV (white arrows). (**C**) Axonin-1-positive commissural neurons, located in the dorsal horn of the spinal cord, carried a primary cilium at HH26 in vivo (white arrows). (**D**) Commissural neurons also possessed a primary cilium in explants cultured for 1 DIV (white arrows). (**E**,**F**) Cultured dissociated motoneurons (**E**) and commissural neurons (**F**), stained for neurofilament (NF-M), carried a primary cilium stained with Arl13B at different levels after 1, 2, 5, and 7 DIV (white arrows). (**G**) About half of the dissociated motoneurons carried a primary cilium after 1 and 2 DIV, and the ciliation rate significantly increased up to around 80% after 5 and 7 DIV. One-way ANOVA with Dunn’s multiple-comparisons test. (**H**) More than 65% of cultured dissociated commissural neurons carried a primary cilium after 1 DIV and kept it to a similar level up to 7 DIV. Kruskal–Wallis test with Dunn’s multiple-comparisons test. Error bars represent standard deviations. *p* ≥ 0.05, ns; *p* < 0.0001, ****. MN, motoneurons; CN, commissural neurons. Scale bars: 10 μm.

**Figure 8 ijms-22-03176-f008:**
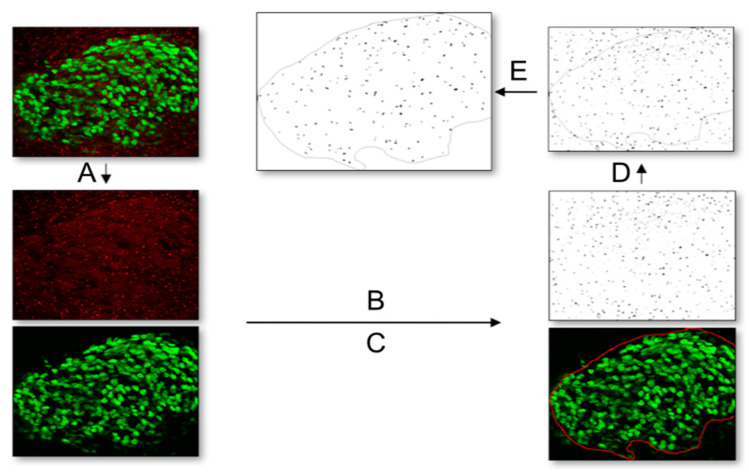
Workflow for quantification of primary cilia with the image analysis program Fiji. (**A**) First, a 2-channel picture with Islet-1 (green) and Arl13B (red) was split into separate channels. (**B**) On the picture displaying the primary cilia (Arl13B, red), the “triangle” threshold was applied. (**C**) The picture showing the Islet-1-positive DRG neurons was used to select the region of interest of the DRG with the freehand selection tool. (**D**) This selection was transferred to the thresholded primary cilia picture. (**E**) Finally, cilia were automatically counted within the region of interest using the “count particles” option.

**Table 1 ijms-22-03176-t001:** Percentage of ciliation of dissociated neurons. Detailed values for Figure 6 and Figure 7.

	DRG				Schwann Cells			
	1 DIV	2 DIV	5 DIV	7 DIV	1 DIV	2 DIV	5 DIV	7 DIV
**average:**	19%	20%	25%	24%	89%	90%	81%	92%
**standard deviation:**	6%	6%	6%	7%	5%	5%	3%	3%
**N(replicates):**	5	5	5	5	3	3	3	3
	**Commissural Neurons**				**Motor Neurons**			
	1 DIV	2 DIV	5 DIV	7 DIV	1 DIV	2 DIV	5 DIV	7 DIV
**average:**	66%	68%	66%	76%	48%	47%	80%	85%
**standard deviation:**	4%	6%	7%	5%	8%	4%	9%	7%
**N(replicates):**	4	4	4	4	6	6	4	4

**Table 2 ijms-22-03176-t002:** List of primary antibodies used for immunohistochemistry and immunocytochemistry.

Antigen	Species	Source	Cat#	Dilution
Arl13B	Rabbit (polyclonal)	Proteintech Group (Manchester, UK)	13967-1-AP, RRID:AB_2121979	1:1000
Ift88	Rabbit (polyclonal)	Proteintech Group	17711-1-AP, RRID:AB_2060867	1:1000
Islet1 (clone 40.2D6)	Mouse (monoclonal)	DSHB (Iowa City, IA, USA)	40.2D6, RRID:AB_528315	1:30 (supernatant)
MelEM	Mouse (monoclonal)	DSHB	MelEM, RRID:AB_531849	1:2 (supernatant)
P0 (clone 1E8)	Mouse (monoclonal)	DSHB	1E8, RRID:AB_2078498	1:2 (supernatant)
HNK1 (clone 1C10)	Mouse (monoclonal)	DSHB	1C10, RRID:AB_10570406	1:2 (supernatant)
Axonin-1/Contactin-2	Goat (polyclonal)	Sonderegger Lab	NA	1:1000
Tyrosine Hydroxylase	Mouse (monoclonal)	DSHB	aTH, RRID:AB_528490	1:5 (supernatant)
Neurofilament-M (clone RMO270)	Mouse (monoclonal)	Invitrogen (Thermo Fisher Scientific, Waltham, MA, USA)	RMO270, RRID:AB_2315286	1:250

**Table 3 ijms-22-03176-t003:** List of secondary antibodies used for immunostaining.

Secondary Antibodies	Source	Cat#	Dilution
Donkey-anti-mouse IgG-Alexa-488	Invitrogen (Thermor Fisher Scientific, Waltham, MA, USA)	A21202 RRID:AB_141607	1:1000
Donkey-anti-rabbit-Cy3	Jackson ImmunoResearch (West Grove, PA, USA)	715-165-152 RRID: AB_2307443	1:1000
Donkey-anti-goat-Cy5	Jackson ImmunoResearch	705-175-147 RRID: AB_2340415	1:1000
Donkey-anti-Goat IgG-Alexa-488	Invitrogen	A11055 RRID:AB_2534102	1:1000

**Table 4 ijms-22-03176-t004:** Media contents for different cell populations in dissociated cell cultures and explants.

Cell Populations				
DRG	MEM with Glutamax (Invitrogen, cat# 41090-028)	Albumax (4 mg/mL, Invitrogen, cat# 11020-021)	N3	NGF (20 ng/mL; Invitrogen, cat# 13290-010)
Commissural neurons/motor neurons	MEM with Glutamax (Invitrogen, cat# 41090-028)	Albumax (4 mg/mL, Invitrogen, cat# 11020-021)	N3	Pyruvate (1 mM; Sigma, cat# P5280)

## Data Availability

Not applicable.

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
