# Peer review of "Investigating Primary Cilia during Peripheral Nervous System Formation"

_ijms, 2021, doi:10.3390/ijms22063176_

Round 1
Reviewer 1 Report
Cilia have been shown to play a major signaling role in various aspects of neural development. In particular, human ciliopathies exhibit several characteristics potentially, or even demonstrated to be associated with abnormalities of neural crest cells. In this manuscript, the authors study the presence of cilia in cells derived from the neural crest of the trunk, at different stages of migration or differentiation in the chicken embryo. By primarily detecting Arl13b, they show the presence of cilia in neural crest cells, sensory neurons and boundary cap cells. In addition, they show that DRG cells lose their cilia when they are dissociated in culture, but not when they are cultivated in explants. The results are convincing and provide a useful description of the presence of cilia in cells derived from the trunk neural crest in the chicken embryo.
I only have a few minor suggestions prior to publication:
It could be fair to cite one of the first article indicating a cell autonomous role of cilia in cranial neural crest cells: Tobin et al. 2008. As well, trunk neural crest cells are involved in heart development and evidences from the cbs mutant suggest a role of cilia, not in their migration, but in their differentiation. This could be cited (Willaredt et al, 2012).
Cilia have been already shown to be present on DRG in mice and in melanocytes in mouse and zebrafish… The authors should more explicitly mention these similar observations in other model organisms. In this regard, a previous review on cilia in NCC could be cited: https://doi.org/10.1016/bs.ctdb.2014.11.004
Author Response
Cilia have been shown to play a major signaling role in various aspects of neural development. In particular, human ciliopathies exhibit several characteristics potentially, or even demonstrated to be associated with abnormalities of neural crest cells. In this manuscript, the authors study the presence of cilia in cells derived from the neural crest of the trunk, at different stages of migration or differentiation in the chicken embryo. By primarily detecting Arl13b, they show the presence of cilia in neural crest cells, sensory neurons and boundary cap cells. In addition, they show that DRG cells lose their cilia when they are dissociated in culture, but not when they are cultivated in explants. The results are convincing and provide a useful description of the presence of cilia in cells derived from the trunk neural crest in the chicken embryo.
We thank Reviewer 1 for the positive assessment of our manuscript.
I only have a few minor suggestions prior to publication:
It could be fair to cite one of the first article indicating a cell autonomous role of cilia in cranial neural crest cells: Tobin et al. 2008. As well, trunk neural crest cells are involved in heart development and evidences from the cbs mutant suggest a role of cilia, not in their migration, but in their differentiation. This could be cited (Willaredt et al, 2012).
We thank Reviewer 1 for suggesting these two papers. We have added the references to the first paragraph of the Discussion (line 256).
Cilia have been already shown to be present on DRG in mice and in melanocytes in mouse and zebrafish… The authors should more explicitly mention these similar observations in other model organisms. In this regard, a previous review on cilia in NCC could be cited: https://doi.org/10.1016/bs.ctdb.2014.11.004
We thank Reviewer 1 to draw our attention on these studies. As the suggested review is published in a hard-to-get journal and because the data are not shown, we opted for another paper: Tan et al. PNAS, 2007 104 (44) 17524-17529; https://doi.org/10.1073/pnas.0706618104
Reviewer 2 Report
The role of primary cilia in PNS development remains poorly understood. In this manuscript, the authors addressed whether primary cilia are present in neural crest cells (NCC) and their derivatives in vivo. They also assessed ciliation of dissociated and explant cultures of neurons in PNS and CNS in vitro and found differences in the capacity for ciliogenesis in vitro among some neuronal cell types once they are dissociated. Overall, this is a descriptive study that lacks functional insights, but provides some useful knowledge for the future functional studies of primary cilia in PNS.
Major Points:
- This study evaluated cilia number of NCC and derivatives during PNS development. Given our knowledge that cilia length and morphology play an important role in regulating ciliary signaling pathways, it is critical to also analyze cilia length quantitatively in parallel with cilia number during NCC migration and differentiation.
- All figures replied on a single cilia marker Arl13B for analysis of ciliation. Some prior studies, such as Ashley Sterpka et al., Pharmacological Research 137 (2018)114-121, suggested that AC3 may be a better marker than Arl13B for neuronal cilia. When assessing cilia in different populations of NCC derivatives, it would be reassuring to confirm some of the key results with a second cilia marker, such as AC3 or IFT88 (as used in Appendix A).
- In-depth analyses of sonic hedgehog signaling pathway in vivo and in vitro (such as nuclear translocation of Gli1/Gli2 and Gli3 processing into repressive forms) would add significance to the current study in terms of characterizing not only cilia number and length but also ciliary signaling in different populations of PNS neurons.
- What implications do the differences in the capacity for ciliogenesis in vitro among different neuronal cell types have in vivo? Please elaborate in the discussion.
Minor Points:
- In Fig. 1, what percentage of NCC+ positive cells have a primary cilium? What is the identity of NCC- negative cell that have a primary cilium? Please confirm results with a different marker, AC3 or IFT88.
- In Fig. 2, what percentage Islet1+ cells have cilia and what percentage Islet- cells have cilia? Is it possible to use a Schwann cell marker to confirm the speculation about the identity of Islet- cells with cilia?
- In Fig. 4, compared with the Islet1 staining, it seems that the TH staining lacks specificity as there is no TH-negative cells.
- In Fig. 5, dissociated DRG neurons showed no cilia by Arl13B. This negative result needs to be confirmed with another cilia marker. Is the non-ciliary diffuse Arl13B staining due to a truncated protein product or mislocalization of the full-length protein? A Western blot may provide some useful information.
- References #24 and #25 are duplicates.
Author Response
The role of primary cilia in PNS development remains poorly understood. In this manuscript, the authors addressed whether primary cilia are present in neural crest cells (NCC) and their derivatives in vivo. They also assessed ciliation of dissociated and explant cultures of neurons in PNS and CNS in vitro and found differences in the capacity for ciliogenesis in vitro among some neuronal cell types once they are dissociated. Overall, this is a descriptive study that lacks functional insights, but provides some useful knowledge for the future functional studies of primary cilia in PNS.
We thank Reviewer 2 for the positive assessment of our manuscript.
Major Points:
This study evaluated cilia number of NCC and derivatives during PNS development. Given our knowledge that cilia length and morphology play an important role in regulating ciliary signaling pathways, it is critical to also analyze cilia length quantitatively in parallel with cilia number during NCC migration and differentiation.
We now have added the quantification of the percentage of ciliation in migrating neural crest cells, boundary cap cells and sympathetic neurons. Moreover, we quantified the average ciliary length for NCC at HH18, as wells as DRG, BCC and SG at HH26. All data are given in the text of the respective result sections.
All figures replied on a single cilia marker Arl13B for analysis of ciliation. Some prior studies, such as Ashley Sterpka et al., Pharmacological Research 137 (2018)114-121, suggested that AC3 may be a better marker than Arl13B for neuronal cilia. When assessing cilia in different populations of NCC derivatives, it would be reassuring to confirm some of the key results with a second cilia marker, such as AC3 or IFT88 (as used in Appendix A).
We thank the Reviewer for bringing up this important point. Indeed, ACIII may be a better marker for neuronal primary cilia. However, the lack of a good commercial antibody that we could use in the chicken precluded such experiment. Nevertheless, we now show IFT88 staining of cilia in SG and BCC at HH26 as well (Supplementary Figure 2 ) to confirm the observation made with Arl13B staining.
In-depth analyses of sonic hedgehog signaling pathway in vivo and in vitro (such as nuclear translocation of Gli1/Gli2 and Gli3 processing into repressive forms) would add significance to the current study in terms of characterizing not only cilia number and length but also ciliary signaling in different populations of PNS neurons.
This kind of experiments would be very interesting to perform. It would give some insights on the possible activation of the sonic hedgehog (Shh) canonical pathway during NCC migration and development of their derivatives. However, this would be beyond the scope of the characterization we have performed in this study. Of course, Shh might play an important role via the primary cilium during PNS formation. However, other important signaling cascades involved in the PNS formation might be transduced at the cilium level as well (e.g. Wnt, TGF-β; see our discussion). Thus, we would not like to focus specifically on any of these at this stage and wish to perform such experiments in the future, when we will characterize the potential function of the primary cilium during PNS formation in detail.
What implications do the differences in the capacity for ciliogenesis in vitro among different neuronal cell types have in vivo? Please elaborate in the discussion.
We thank Reviewer 2 for bring up this interesting point. We have now elaborated this point in the discussion with some parallels to regenerative properties of PNS neurons versus CNS neurons (lines 328-337).
Minor Points:
In Fig. 1, what percentage of NCC+ positive cells have a primary cilium? What is the identity of NCC- negative cell that have a primary cilium? Please confirm results with a different marker, AC3 or IFT88.
We have now quantified the percentage of Arl13B-positive cilia in migrating NCC cells at HH18. As mentioned above, we cannot perform AC3 staining in the chicken due to the lack of a good antibody. IFT88 can be used to assess ciliation at a qualitative level. However, for percentage of ciliation and ciliary length measurement it is not adequate, because it stains not only the axoneme but also the basal body in ciliated cells and the basal body in non-ciliated cells (Robert et al., 2007) which would lead to counting false-positive cells. Arl13B stains only the axoneme and is therefore the best marker that we can use in the chicken (Caspary et al., 2007). We have expanded our confirmatory staining with anti-IFT88 antibodies to boundary cap cells and sympathetic neurons.
Cells in the vicinity of migrating NCC are most likely cells from the dermomyotome or sclerotome depending on the localization in the dorso-ventral axis. However, due to the lack of adequate markers and because of our focus on neural crest cells, we did not identify cells in the vicinity of neural crest cells.
In Fig. 2, what percentage Islet1+ cells have cilia and what percentage Islet- cells have cilia? Is it possible to use a Schwann cell marker to confirm the speculation about the identity of Islet- cells with cilia?
We give the percentage of ciliation in DRG (see paragraph 2.2). We could not give the percentage of Islet1+ cells that bore a primary cilium, because it is very often difficult to clearly identify the cell soma (nucleus) to which a cilium belongs, as the cell populations in DRG are heterogeneous. We can only indirectly quantify that the large majority of Islet1-positive neurons is ciliated because they constitute the large majority of cells in DRG at HH26 (76%).
We now have added 1E8 staining of Schwann cells precursor in HH26 DRG together with Arl13B or IFT88 co-staining showing their ciliation in vivo (Supplementary Figure 2).
In Fig. 4, compared with the Islet1 staining, it seems that the TH staining lacks specificity as there is no TH-negative cells.
Islet-1 is a nuclear marker whereas TH is in the cytoplasm of sympathetic neurons. For that reason, the staining looks very different from Islet1 because also cytoplasmic protrusions are detected. As they might grow in-between cells that are not SG neurons in seems that all cells are positive. Thus, we don’t think that it is a lack of specificity of this antibody to SG but that it is just difficult/impossible to see single cells with such a staining. We used it to verify that these cells are sympathetic neurons and as an complementary staining to Islet-1, which is the best marker for such an experiment.
In Fig. 5, dissociated DRG neurons showed no cilia by Arl13B. This negative result needs to be confirmed with another cilia marker. Is the non-ciliary diffuse Arl13B staining due to a truncated protein product or mislocalization of the full-length protein? A Western blot may provide some useful information.
As mentioned above the only other ciliary marker we could use for such an experiment would be Ift88. Unfortunately, so far we do not have any antibody for any other primary cilium-specific marker that work in the chicken beside this one. Unfortunately, Ift88 also stains the basal body of non-ciliated cells as mentioned above. This would not be useful to rule out whether the staining reveals a cilium or a basal body and would lead to false positives.
However, because we were able to detect primary cilia in DRG neurons in vivo and when cultured as explants, we consider it very unlikely that the absence of a primary cilium in dissociated DRG neurons would be due to a problem with the antibody that would not recognize Arl13B under these conditions.
References #24 and #25 are duplicates.
We apologize for this mistake. This has been corrected.
Round 2
Reviewer 2 Report
In this revision, the authors have addressed most of my original comments satisfactorily with further experimentation or clarification.
The presentation of results can be improved by presenting the quantification data of ciliation and cilia length in figures, rather than only showing the numbers in the text. Please see details below.
Specific Points:
Original Question: This study evaluated cilia number of NCC and derivatives during PNS development. Given our knowledge that cilia length and morphology play an important role in regulating ciliary signaling pathways, it is critical to also analyze cilia length quantitatively in parallel with cilia number during NCC migration and differentiation.
Author’s Response: We now have added the quantification of the percentage of ciliation in migrating neural crest cells, boundary cap cells and sympathetic neurons. Moreover, we quantified the average ciliary length for NCC at HH18, as wells as DRG, BCC and SG at HH26. All data are given in the text of the respective result sections.
Reviewer’s Response: Quantification data about ciliation and cilia length are key parameters for not only assessing cilia formation and morphology but also implicating cilia function, therefore these data should be summarized in figures, not just given in the text, similar to what the authors showed in Fig. 5 and 6.
Original Question: In-depth analyses of sonic hedgehog signaling pathway in vivo and in vitro (such as nuclear translocation of Gli1/Gli2 and Gli3 processing into repressive forms) would add significance to the current study in terms of characterizing not only cilia number and length but also ciliary signaling in different populations of PNS neurons.
Author’s Response: This kind of experiments would be very interesting to perform. It would give some insights on the possible activation of the sonic hedgehog (Shh) canonical pathway during NCC migration and development of their derivatives. However, this would be beyond the scope of the characterization we have performed in this study. Of course, Shh might play an important role via the primary cilium during PNS formation. However, other important signaling cascades involved in the PNS formation might be transduced at the cilium level as well (e.g. Wnt, TGF-β; see our discussion). Thus, we would not like to focus specifically on any of these at this stage and wish to perform such experiments in the future, when we will characterize the potential function of the primary cilium during PNS formation in detail.
Reviewer’s Response: Given the dependence of Shh signaling on the primary cilium, an evaluation of Shh alone is still very meaningful in terms of providing some critical functional insights. It is a fair point made by the authors that a more comprehensive analysis of ciliary signaling pathways can be conducted when addressing functions of PC in PNS formation in future studies.
Minor point:
In method section 4.4: Importantly, to stain primary cilia in the PNS, no detergent was used and primary cilia were stained with rabbit anti-Arl13B antibody (Proteintech) diluted at 1:1000 in 5% FCS in PBS for 2 h at room temperature (Table 3).
Why the absence of detergent in the buffer is important for staining primary cilia in the PNS? Please elaborate; it would help the readers who are interested in studying PC in PNS.
Author Response
Dear editors,
we are happy to hear that our manuscript has been evaluated positively by both reviewers. While reviewer 1 is happy with the revised version, reviewer 2 asked for a different way of showing our data. We have done this as suggested. We also gave more details on the changes in the staining protocol for cilia. A point-to-point response to reviewer 2's requests is given in the attachment.
Best regards
Esther Stoeckli
